# Multidimensional correlates of psychological stress: Insights from traditional statistical approaches and machine learning using a nationally representative Canadian sample

Benjamin A. Hives[1]*, Mark R. Beauchamp[1], Yan Liu[2¤a], Jordan Weiss[3¤b], Eli Puterman[1]

**1** School of Kinesiology, University of British Columbia, Vancouver, British Columbia, Canada,
**2** Department of Educational and Counselling Psychology, University of British Columbia, Vancouver, British Columbia, Canada, **3** Center on Longevity, Stanford University, Stanford, California, United States of America.

¤aCurrent Address: Department of Psychology, Carleton University, Ottawa, Ontario, Canada.
¤bCurrent Address: Department of Medicine, New York University, New York, New York, United States of America.
* ben.hives@ubc.ca

## Abstract

Approximately one-fifth of Canadians report high levels of psychological stress. This is alarming, as chronic stress is associated with non-communicable diseases and premature mortality. In order to create effective interventions and public policy for stress reduction, factors associated with stress must be identified and understood. We analyzed data from the 2012 'Canadian Community Health Survey - Mental Health' (CCHS-MH), including 66 potential correlates, drawn from a range of domains (e.g., psychological, physical, social, demographic factors). First, we used a random forest algorithm to determine the most important predictors of psychological stress, then we used linear regressions to quantify the linear associations between the important predictors and psychological stress. In total, 23,089 Canadian adults responded to the 2012 CCHS-MH, which was weighted to be nationally representative. Random forest analyses found that, after accounting for variance from other factors and considering complex interactions, life satisfaction (relative importance = 1.00), negative social interactions (0.99), primary stress source (0.85), and age (0.77) were the most important correlates of psychological stress. To a lesser extent, employment status (0.36), was also an important variable. Univariable linear regression suggested that these variables had effects ranging from small to medium-to-large. Multiple linear regression showed that lower life satisfaction, being younger, greater negative social interaction, reporting a primary stressor, and being employed were all found to be associated with greater psychological stress (beta range = 0.03 to 0.84, all $p < 0.001$, $R^2 = 0.264$). Further, these factors accounted for 26% of the variance of psychological stress. This study highlights that the most important correlates of stress reflect diverse psychological, social,

**Data availability statement:** The data underlying the results presented in the study are available on Borealis (https://borealisdata.ca/dataset.xhtml?persistentId=doi:10.5683/SP3/AKDVSL).

**Funding:** The author(s) received no specific funding for this work.

**Competing interests:** The authors have declared that no competing interests exist.

and demographic factors. These findings highlight that stress reduction interventions may require a multidisciplinary approach. However, further longitudinal and experimental studies are required.

## Introduction

In Canada, more than one-fifth of adults admitted most days in their daily lives to be "quite a bit" or "extremely" stressful [1]. This is alarming considering the serious health consequences of chronic psychological stress [2–4]. Chronic psychological stress is well-evidenced to contribute to poor health behaviours (e.g., smoking, poor diet, physical inactivity) [5], as well as the development of psychiatric problems (e.g., depression) [6] and chronic diseases (e.g., cardiovascular disease, cancer) [3]. The deleterious effects of stress are recognized by several government-funded organizations that have incorporated stress reduction in their messaging to the public. Health Canada [7] has noted that high psychological stress is a risk factor for developing heart disease, mental illness, and bowel disease, similar to warnings from the American Heart Association [8]. The World Health Organization [9] warns that stress may not just lead to physical symptoms, but also emotional symptoms and behavioural changes (e.g., difficulty concentrating, poor eating and sleeping behaviours).

While the long-term consequences of stress have been well-defined, the antecedents and sources of chronic psychological stress are complex and derive from multiple biological and socio-ecological domains (e.g., genetic make-up [10], lifetime or recent exposure to stressors [11,12], personality [13], and health behaviours [2,14]). For example, high stress is more commonly reported by females/women compared to males/men [15,16], by younger compared to middle-aged adults [15], by ethnic minorities [17], and by those with lower education and incomes [18]. However, many of these findings stem from studies with *a priori* hypotheses which examine predictors of psychological stress in relative isolation. While hypothesis testing frameworks are the foundation of the scientific method, it is difficult to create separate hypotheses to account for complex interactions between a large volume of concurrent correlates, which may inhibit a holistic understanding of the most salient factors that are associated with perceptions of psychological stress. Further, relying on an *a priori* framework limits the breadth of predictors considered, which can lead to overlooking unanticipated factors.

In addition to these framework considerations, the majority of past analyses have relied exclusively on traditional statistical methods (e.g., regression). Traditional statistical methods are not able to account for numerous variables due to multicollinearity nor the possible complex interactions among correlates without violating the underlying assumptions of regression. Often, an individual's level of psychological stress is not associated with a singular event or risk factor, but rather is associated with interactions among an array of factors [11]. Machine learning has the potential to incorporate many variables without risk of multicollinearity, while also accounting for complex interactions. This allows researchers to incorporate a greater number of variables from diverse biological and socio-ecological domains, including those that

might previously have been ignored, while modeling complex interactions. In the current analysis, we applied a machine learning approach, namely random forest analysis, which accounts for linear and non-linear effects, as well as complex interactions. However, random forest analyses are unable to isolate direct effects from indirect effects. Rather, it produces a 'variable importance' to represent the combined direct and indirect effects. Therefore, we also incorporated linear regression into our analyses to quantify the direct effects of important variables.

Given that many potential correlates of stress remain unexplored, and those that are examined are usually studied in relative isolation, the objective of this exploratory data analysis was to simultaneously analyze 66 clinical, psychological, social, behavioural, and demographic variables to determine the most important correlates of global perceptions of psychological stress. As this is a data-driven study, there were no *a priori* hypotheses.

## Methods

### Data source

This cross-sectional observational study used the Public Use Microdata File data from the 2012 Canadian Community Health Survey - Mental Health (CCHS-MH) [19]. This survey sampled Canadians aged 15 or older in the 10 provinces and excluded institutionalized populations, individuals living on First Nation reservations and Crown land, individuals living in the three territories, and members of the Canadian Armed Forces. The survey response rate was 68.9% (n = 25,113) and included survey weights to adjust the sample to be representative of the Canadian population [19]. The present analyses used a subsample of the CCHS-MH respondents who were aged 20 years or older (n = 23,089) at the time of survey completion. Data were accessed on April 11, 2018. As the data used in this study are publicly available and fully anonymized, ethics approval was not required.

### Outcome measure

The CCHS-MH measured psychological stress through one item. This item asks "Thinking about the amount of stress in your life, would you say that most days are...?" and includes the Likert-type responses: 'not at all stressful' (1), 'not very stressful' (2), 'a bit stressful' (3), 'quite a bit stressful' (4), and 'extremely stressful' (5). Because this single-item measure contained 5 categories and was approximately normal in its distribution, it was treated as continuous [20].

### Correlates

The 2012 CCHS-MH included a total of 586 variables. Following a review of all variables, two of the authors (BAH, EP) removed variables that had: (1) little or no variation (e.g., less than 1% of the sample reported being pregnant), (2) high theoretical overlap or correlation with other variables (e.g., the social provision scale total score was used rather than each individual item of the scale) or (3) no theoretical connection with stress (e.g., height). In total, we included 66 potential correlates in the present analysis. These variables measure demographic information (n = 14), health behaviours (n = 7), social factors (n = 7), early and recent life adversity (n = 5), mental health indicators (n = 10), psychological factors (n = 7), and physical health indicators (n = 16). See Table 1 for a complete list of variables analyzed and Supplemental S1 Table for a detailed description of each variable used.

### Statistical analyses

**Data processing.** We included and operationalized 66 factors from the dataset. Prior to analyses, all responses of 'don't know', 'refusal', 'not stated', and 'not applicable' (NA) were treated as missing values. Then, all continuous and ordinal variables (e.g., age, social provisions) were converted to standardized scores with mean of 0 and standard deviation of 1. Nominal categorical variables (e.g., primary source of stress, education) were converted to factor-type data, which allows the algorithm to treat them as dummy coded without artificially increasing the likelihood of being predicted

**Table 1. Variables included in the random forest analysis, by category.**

| Demographic Factors (14) | | |
|---|---|---|
| • Age<br>• Body Mass Index<br>• Dwelling type<br>• Education<br>• Employment | • Household size<br>• Household type<br>• Immigrant status<br>• Income<br>• Marital status | • Province of residence<br>• Sex<br>• Student status<br>• Visible minority status |
| **Health Behaviours (7)** | | |
| • Alcohol abuse or dependence<br>• Drug abuse or dependence<br>• Frequency of drinking | • Illicit drug use<br>• Level of insomnia | • Smoking status<br>• Weekly hours of moderate or vigorous physical activity |
| **Life Adversity (5)** | | |
| • Early life adversity - Sum Score<br>• Recent life events - Family problems | • Recent life events - Unmet needs<br>• Recent life events - Victim of a crime | • Recent life events - Witness a crime |
| **Psychological Factors (7)** | | |
| • Coping on a daily basis<br>• Coping skill<br>• Coping with crisis | • Difficulty concentrating<br>• Emotional impact of health | • Greatest source of stress<br>• Life satisfaction |
| **Social Factors (7)** | | |
| • Community belonging<br>• Coping - social support<br>• Difficulty in community activities | • Difficulty maintaining friendship<br>• Difficulty with new people | • Negative social interactions<br>• Social provisions |
| **Mental Health (10)** | | |
| • Anxiety disorder<br>• Attention deficit disorder<br>• Bipolar disorder<br>• Generalized anxiety disorder | • Hypomania<br>• Learning disability<br>• Major depressive episode | • Mania<br>• Post-traumatic stress disorder<br>• Suicidal thoughts |
| **Physical Health (16)** | | |
| • Arthritis<br>• Asthma<br>• Back problems<br>• Bowel disorders<br>• Chronic fatigue<br>• Current cancer | • Diabetes<br>• Difficulty with household responsibilities<br>• Difficulty standing<br>• Difficulty walking<br>• Heart disease | • High blood pressure<br>• Migraines<br>• Previous cancer<br>• Self-perceived health<br>• Stroke |

in the random forest (e.g., if smoking status was converted to 6 binary variables, it would be six times more likely to be selected at each split compared to a non-dummy coded variable). We used a random forest-based method to impute missing values based on a participant's responses in all other variables. We coded numeric variables such that higher scores indicated a positive association with stress.

**Random forest analysis.** The primary analysis for this study was random forest analysis [21], a decision tree-based machine learning method. Consistent with standard machine learning practices, the dataset was split into two subsets, a training set (75% of observations) for model development and a testing set (25% of observations) for evaluating the model's performance with new data. To ensure similar distributions of the outcome variable in both datasets, stratified sampling (strata) was applied at this initial data splitting, which segmented the data of the outcome variable into quantiles and randomly sampled from each. To further assess model performance, we employed 10-fold cross-validation to evaluate the performance of a Random Forest model by partitioning the dataset into ten equal subsets, or 'folds'. In each iteration, the model was trained on nine folds and validated on the remaining fold, ensuring that every observation was used for both training and validation exactly once. This process helps assess the model's generalizability and reduces the risk of overfitting.

 

Additionally, we performed hyperparameter tuning, using an automated search of 11 different hyperparameter combinations, including the number of variables tested at each split and the minimum number of observations required in a terminal node. The values for these two hyperparameters were automatically sampled from reasonable default ranges defined by the R package used in the analysis. The optimal model was selected based on the lowest mean squared error (MSE) obtained from cross-validation. Hyperparameter tuning helps further optimize model performance by improving predictive accuracy and reducing error rates and enhances generalization by increasing the model's ability to perform well on new or unseen data.

Random forest analysis was chosen as it captures complex interactions and nonlinear patterns [22]. Additionally, it can provide a metric for variable importance, known as feature importance in machine learning, which is a measure of the extent to which model error changes if a variable is replaced with a randomly permutated set of numbers [22,23]. In other words, high variable importance represents a variable having a large direct and/or indirect effect on the outcome through appearing more often and at earlier (i.e., higher) splits. In this study, variable importance is noted in terms of absolute value (labeled "importance") or as a proportion of the greatest importance value (labeled "relative importance").

**Linear regression.** Using the predictors with highest importance from the random forest analysis, we first built simple linear regression models with each important variable, then we built a multiple linear regression model with all important variables to quantify the direct associations and overall variance explained, respectively.

**Sensitivity analysis.** Data for coping skills, coping - social support, and previous year employment were missing by design; that is, the missing values were a function of survey design (i.e., coping items were not asked to those who did not report a greatest source of stress, and employment items were not asked of those older than 75). Sensitivity analyses were used to determine if the results were robust against these missing data, following the same procedure as our primary analysis but using different subsets of the data. First, an *age-restricted analysis* included only participants aged 20–75 years to address employment-related missing data. Second, a *coping-excluded analysis* omitted coping skill and coping - social support variables. Third, a *mental health-restricted analysis* included only participants without current mental illness to ensure results were not driven by individuals with severe mental health conditions. In this analysis, we excluded those with major depressive disorder, bipolar disorder, PTSD, suicidal thoughts, anxiety disorders, mania, or hypomania.

To compare these analyses to the primary analysis, we used three common error measures (i.e., MSE, root mean squared error, mean absolute error) and one effect size (i.e., $R^2$)[24]. MSE is the average of squared differences between predicted and actual data, while root mean squared error (RMSE), the square root of the MSE, provides an error measure on the same scale as the original data (i.e., stress score). Mean absolute error (MAE) is the average of the absolute differences between predicted and actual data, which is less influenced by outliers. $R^2$ represents the proportion of the variance in the outcome variables explained by the model, assessing an overall model fit. Consistency between these sensitivity analyses and the primary analysis would confirm the robustness of the findings, demonstrating that the results were not unduly influenced by missing data or the inclusion of participants with mental illness.

**Software.** R statistical software was used to conduct all analyses [25]. Functions from the *tidyverse* package were used for the data preprocessing and visualizations [26]. The *missRanger* package was used to conduct the random forest imputation [27]. The *lm* function from the *stats* package was used for the linear regressions with survey weights [25] and the *emmeans* package [28] was used to conduct a post-hoc test to find differences in the stress source linear model. The random forest analysis was conducted using *ranger* [29] to create the random forest model, *vip* [30] to extract the variable importance, and functions from the *tidymodels* [31], *usemodels* [32], and *textrecipes* [33] to create a tidymodel framework.

## Results

### Demographic factors

The weighted sample ($N_{raw}$ = 23,089; $N_{Weighted}$ = 26,020,507) was composed of a slight majority of female participants (51%). Sixty-five percent of the sample was married or common-law. Nearly one-quarter (22%) identified as a visible minority.

Nearly two-thirds had completed post-secondary education (63%). The median age bracket was 45–49. Additional demographic information can be found in Supplemental S2 Table – Weighted Sample Demographic Information.

## Random forest

The random forest model included 1000 trees, as suggested by Breiman [34] for analysis interested in variable importance. Our final model included 21 variables at each split and had a minimum node size of ten. The model was able to account for 31% of the variance in psychological stress ($R^2 = 0.31$). The variables with the greatest relative importance were life satisfaction (relative importance = 1.00), frequency of negative social interactions (0.99), primary stress source (0.85), and age (0.77). The variable with the next greatest importance was the previous year employment status (0.36). All other variables had a relative importance of less than 0.25, representing less than one-quarter as important as the most important variable. Full variable importance is presented in supplemental materials – S3 Table and S1 Fig.

## Linear regression

**Univariable linear regression.** In five independent univariable linear regression models, each including one of the top five predictors identified by the random forest model, the explained variance in psychological stress ranged from 4% to 14%. Less life satisfaction, more frequent negative social interactions, younger age, and being employed over the previous year were all significantly associated with greater psychological stress. In terms of primary stress sources, those who selected that they had no source of stress were associated with lower levels of psychological stress when compared to participants who reported that they could identify a primary stress source (i.e., estimated marginal means). Potential sources of stress ranged from within a broad range of life domains, including work, personal relationships, school, finances, time pressure, employment status, caring for others and children, family responsibility, and mental and physical health. The univariable regression results are displayed in Table 2 and the estimated marginal means are shown in Fig 1.

## Multiple linear regression

The use of multiple regression allows for an assessment of each predictor, after holding all others consistent. In terms of significance and direction of effects, the multiple linear regression demonstrated similar results to the univariable linear regression. Accounting for the direct effects of all important variables, there was a large effect size (*Adj. $R^2$ = 0.264*). See Table 3 for the full multiple linear regression results.

## Missing data

The vast majority of variables (n = 63/66; 95%) were missing fewer than 7.5% of observations, which is below the 10% threshold to be considered to add bias when data are not missing at random [35]. See Supplemental Table S3 – Missingness of Data for individual variables' levels of missingness. The three variables missing more than 10% of observations were Coping - Social Support (16%), Coping Skill (16%), and Employment (12%), which were missing due to survey design logic (these were only presented to respondents if participants reported a greatest source of stress and were under the age of 76, respectively). To ensure that the analyses were not biased by variables with the greatest amounts of missingness, sensitivity analyses were conducted and are presented below.

## Sensitivity analyses

All sensitivity analyses showed similar overall error (mean squared error range = 0.688–0.701) and explanatory power ($R^2$ range: 0.29–0.31) compared to the primary model (mean squared error = 0.70; $R^2$ = 0.31). Measures of error and $R^2$ are provided in Table 4. All models (primary analysis and sensitivity analyses) identified life satisfaction, negative social

**Table 2.** Univariable linear regression including the most important predictors of psychological stress. Each variable represents a separate regression analysis. Beta estimate, standard error (SE), 95% confidence intervals (CI), and p value of the variable of interest, as well as the $R^2$ of the entire model are included below. See Supplemental S1 File for complete regression results.

| | Estimate | SE | 95% CI | p | $R^2$ |
|---|---|---|---|---|---|
| Life Satisfaction (R) | 0.317 | 0.007 | [0.30, 0.33] | < 0.001 | 0.09 |
| Negative Social Interactions | 0.356 | 0.006 | [0.34, 0.37] | < 0.001 | 0.13 |
| Age (R) | 0.238 | 0.008 | [0.22, 0.25] | < 0.001 | 0.04 |
| Employment | 0.452 | 0.014 | [0.42, 0.48] | < 0.001 | 0.04 |
| Stress Source<br>*Ref Group: "Nothing"* | | | | | 0.14 |
| Caring for own children | 1.017 | 0.038 | [0.94, 1.09] | < 0.001 | |
| Caring for others | 1.068 | 0.060 | [0.95, 1.18] | < 0.001 | |
| Discrimination | 0.809 | 0.140 | [0.53, 1.08] | < 0.001 | |
| Employment status | 1.069 | 0.042 | [0.99, 1.15] | < 0.001 | |
| Financial situation | 1.106 | 0.023 | [1.06, 1.15] | < 0.001 | |
| Health of family members | 0.870 | 0.029 | [0.81, 0.93] | < 0.001 | |
| Loss of loved one | 0.861 | 0.128 | [0.61, 1.11] | < 0.001 | |
| Own emotional or mental health problem | 1.238 | 0.043 | [1.15, 1.32] | < 0.001 | |
| Other | 0.752 | 0.035 | [0.68, 0.82] | < 0.001 | |
| Personal relationships | 1.126 | 0.032 | [1.06, 1.19] | < 0.001 | |
| Other personal or family responsibility | 0.950 | 0.031 | [0.89, 1.01] | < 0.001 | |
| Personal and family's safety | 0.794 | 0.046 | [0.70, 0.88] | < 0.001 | |
| Physical health | 0.899 | 0.030 | [0.84, 0.96] | < 0.001 | |
| School | 1.107 | 0.042 | [1.02, 1.19] | < 0.001 | |
| Time pressures / not enough time | 1.069 | 0.027 | [1.02, 1.12] | < 0.001 | |
| Own work situation | 1.210 | 0.021 | [1.17, 1.25] | < 0.001 | |

(R) denotes reverse-coded variables.

interactions, stress source, and age as the four most important variables. Complete results from the sensitivity analyses can be found in the supplemental materials, Table S4 - Sensitivity Analyses.

## Discussion

Using machine learning, we were able to highlight the most important correlates of psychological stress, accounting for complex interactions and non-linear relationships. Our algorithm identified four important correlates, particularly life satisfaction, negative social interactions, reporting that their stress has a primary source from within an identified life domain, and age.

The present analyses found that life satisfaction was the most important correlate of stress, with life satisfaction being negatively associated with stress and having a medium-sized effect. This aligns with several previous studies that explored the relationship between psychological stress and life satisfaction [36–38]. In a study of more than 340,000 individuals, Strine and colleagues [36] found that, compared to those who were very satisfied, those who reported being satisfied or dissatisfied/very dissatisfied with life were more likely to report being distressed. Smyth and colleagues [38] noted that this relationship also holds over multiple time points. Using ecological momentary assessments, they showed that those with high levels of life satisfaction consistently reported lower levels of perceived stress over three consecutive days. Overall, there appears to be a negative relationship between life satisfaction and psychological stress [36,38], but the nature of directionality remains unclear. It remains unknown if being satisfied with one's life has a protective effect against perceptions of stress, or if psychological stress may affect one's overall satisfaction with life, or whether both are

 

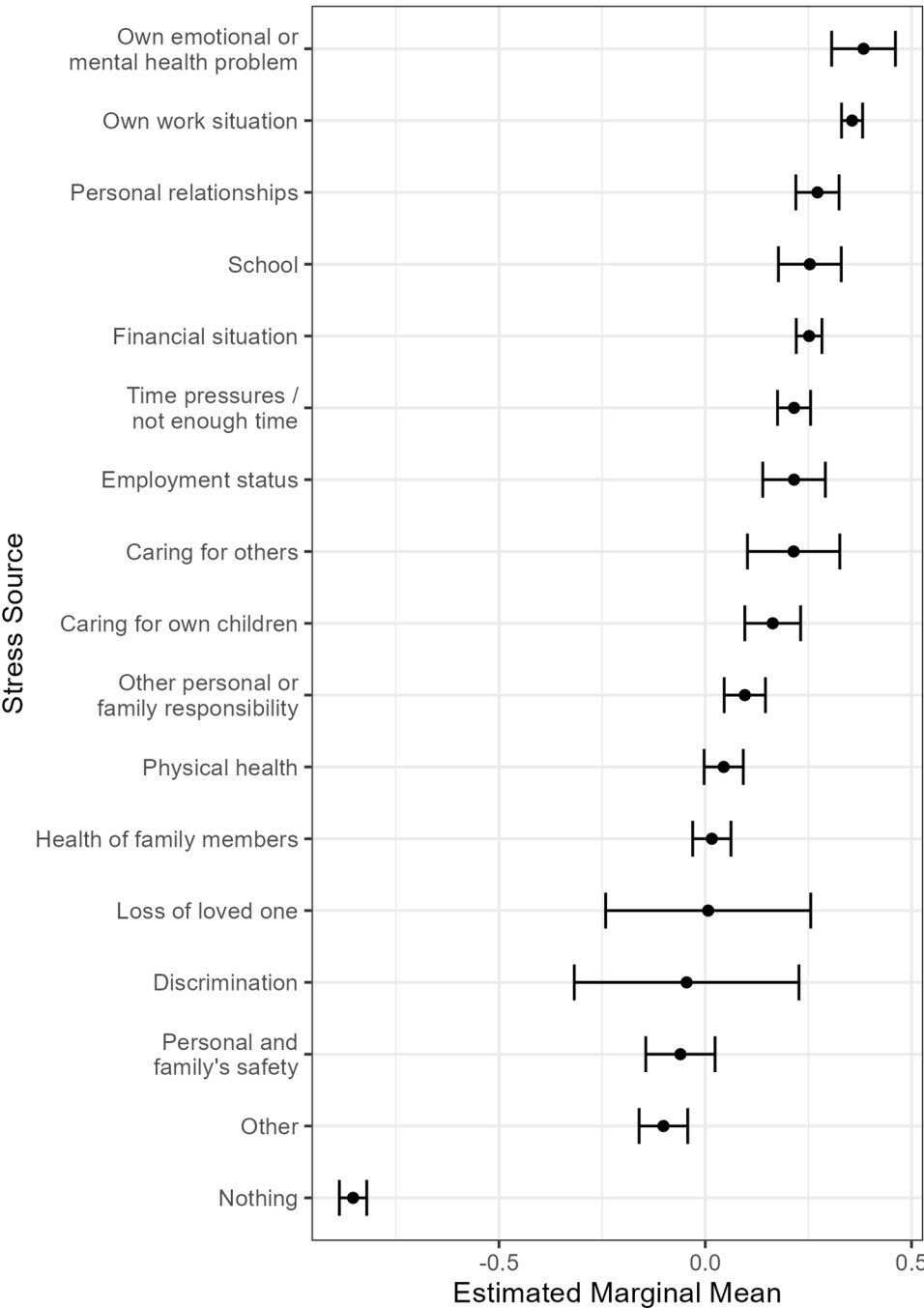

**Fig 1. Estimated marginal means of stress source.**

driven by dispositional attributes (e.g., neuroticism, rumination), or more structural factors (e.g., racism, neighborhood conditions) that were not measured in the CCHS-MH.

In addition to life satisfaction, negative social interactions were found to be another highly important correlate of psychological stress. Negative social interactions demonstrated a significant, medium-to-large, linear positive relationship with

**Table 3. Multiple linear regression including the most important predictors of psychological stress. Beta estimate, standard error (SE), 95% confidence intervals (95% CI), and p value of the variable of interest as well as the intercept are included below. Adj R² represents the adjusted effect size of the entire model.**

| | Estimate | SE | 95% CI | p value |
|---|---|---|---|---|
| (Intercept) | -0.704 | 0.019 | [-0.740, -0.670] | < 0.001 |
| Life Satisfaction (R) | 0.235 | 0.006 | [0.220, 0.250] | < 0.001 |
| Negative Social Interactions | 0.213 | 0.006 | [0.200, 0.230] | < 0.001 |
| Age (R) | 0.034 | 0.008 | [0.020, 0.050] | < 0.001 |
| Employment | 0.198 | 0.015 | [0.170, 0.230] | < 0.001 |
| Stress Source<br>*Ref Group: "Nothing"* | | | | |
| Caring for own children | 0.719 | 0.036 | [0.650, 0.790] | < 0.001 |
| Caring for others | 0.756 | 0.056 | [0.650, 0.870] | < 0.001 |
| Discrimination | 0.521 | 0.130 | [0.270, 0.780] | < 0.001 |
| Employment status | 0.720 | 0.040 | [0.640, 0.800] | < 0.001 |
| Financial situation | 0.703 | 0.023 | [0.660, 0.750] | < 0.001 |
| Health of family members | 0.673 | 0.027 | [0.620, 0.730] | < 0.001 |
| Loss of loved one | 0.668 | 0.119 | [0.440, 0.900] | < 0.001 |
| Own emotional or mental health problem | 0.773 | 0.041 | [0.690, 0.850] | < 0.001 |
| Other | 0.509 | 0.032 | [0.450, 0.570] | < 0.001 |
| Personal relationships | 0.656 | 0.030 | [0.600, 0.720] | < 0.001 |
| Other personal or family responsibility | 0.662 | 0.029 | [0.610, 0.720] | < 0.001 |
| Personal and family's safety | 0.633 | 0.043 | [0.550, 0.720] | < 0.001 |
| Physical health | 0.634 | 0.028 | [0.580, 0.690] | < 0.001 |
| School | 0.782 | 0.041 | [0.700, 0.860] | < 0.001 |
| Time pressures / not enough time | 0.767 | 0.026 | [0.720, 0.820] | < 0.001 |
| Own work situation | 0.838 | 0.022 | [0.800, 0.880] | < 0.001 |
| *Adj R²=0.264* | | | | |

(R) denotes reverse-coded variables.

**Table 4. Model fit comparisons for primary model and sensitivity analyses.**

| | Primary Model (N = 23,089) | Age (20–75) (N = 22,346) | No Coping (N = 21,171) | No Mental Illness (N = 19,775) |
|---|---|---|---|---|
| **MSE** | 0.700 | 0.695 | 0.701 | 0.688 |
| **RMSE** | 0.837 | 0.834 | 0.837 | 0.829 |
| **MAE** | 0.664 | 0.656 | 0.665 | 0.660 |
| **R2** | 0.31 | 0.29 | 0.31 | 0.29 |

MSE, mean squared error; RMSE, root mean squared error; MAE, mean absolute error.

stress. As noted by Cohen, negative social interactions (e.g., arguments with family) are themselves a source of stress [39]. Almeida [40] noted that of all daily stressors reported, half were classified as interpersonal tensions. In addition to negative social interactions being a source of psychological stress, high levels of stress have also been linked to worse social interactions with those closest to an individual [41]. While the causal mechanisms remain unexplored, there is the potential of a bidirectional feedback loop.

The third variable with high importance associated with greater perceptions of psychological stress was the identification of a life domain as a primary source of stress. This was true for any identified life domain, including, but not limited to, personal relationships, caregiving responsibilities, school or work, finances, or their mental or physical health. This is not surprising since global perceptions of psychological stress are, partially, a function of real-life stressors [42]. However, it is surprising that identification of a primary source of stress (compared to not selecting a primary source of their stress) only accounted for 14% of the variation in global psychological stress in the regression analysis. Stress within one life domain has been shown to co-occur with or spillover into other life domains, known as 'stress contagion' [43]. Thus, it is possible that requiring participants to only select one life domain as the primary source missed a substantial amount of overlap with other domains, which, if combined, would have potentially accounted for more variation in psychological stress.

The fourth variable consistently within the top tier of importance was age. Regression analyses showed that there was a negative linear association in which stress was lower for older individuals. Previous studies have noted a similar negative relationship between age and psychological stress [18,44,45]. However, the mechanisms of effect between age and psychological stress remain unclear. Research has previously shown that older adults report fewer exposures to stressors than younger adults, and have lower reactivity to a given stressor [46,47]. This could also be due to older adults reporting lower levels of negative affect and higher levels of positive affect [48]. These improved dispositions may affect older adults' views on potentially stressful situations. For example, previous research has shown age to be negatively associated with negative social interactions [47] and postively associated with life satisfaction [49], which could lead to an indirect effect on psychological stress. Finally, the difference in ranking between the small effect size and one of the largest relative importance values may also be due to the shape of the relationship between age and psychological stress. While some researchers describe this as a linear relationship, work by Stone and colleagues has shown that the relationship between reported daily stressors and age has an inverted-U trend in which daily stress slightly increases during one's 20s and 30s followed by a slow decline until age 50, at which point a dramatic decline occurs [45].

While the present study highlights the complex interactions between predictors of stress in Canadians, it is not without limitations. One limitation is that the data were collected from a single country. While this study highlights the important associates of stress in a Canadians, is in unclear how well it will generalize to countries with differing cultural values. Therefore, it is not clear whether the relationship would be consistent with participants from other countries, when assessing the same factors. Future studies should be completed with large datasets from other countries (e.g., China Health and Retirement Longitudinal Study, Longitudinal Study of Aging in India) to examine if the important correlates of psychological stress are consistent. Another limitation is that this project was limited in the selection of variables by what was present in the CCHS-MH. Future studies would benefit from including variables that have been previously associated with stress (e.g., self-efficacy) [50], but were not included in the CCHS-MH. Finally, there are two limitations regarding our measure of psychological stress (i.e., the Statistics Canada single-item stress measure). First, the measure of psychological stress used by Statistics Canada has not been validated. While it has been used by Statistics Canada and other researchers using Statistics Canada data [44–46] as a measure of psychological stress, future validation research is required to ensure that this measure is assessing an individual's levels of psychological stress, rather than other mental ill-being measures. The second limitation of the single-item measure is the response scale, which is a 5-point Likert-type response scale. Treating Likert-type responses as continuous data is a contentious issue, with some researchers contending it should be treated as ordinal and others contending it can be treated as continuous [51]. Given the robustness of the analyses used (i.e., linear regression models, random forest algorithm), as well as the approximately normal distribution of the responses to the psychology stress item, we decided to treat it as continuous. However, future research should attempt to replicate this study using a measure of psychological stress (e.g., Perceived Stress Scale [47]) with established validity evidence.

## Conclusions

Stress has been shown to have numerous deleterious effects on health and well-being, including increased risk of psychological disease, poor health behaviours, and premature mortality. In order to help combat the high rate of stress in Canadians, the most important predictors of stress must be identified. However, there currently is a paucity of multidisciplinary research that highlights the variables that are the most important. In the present study, we used random forest algorithms with data collected from the Canadian Community Health Survey - Mental Health. We found that while many factors are associated with psychological stress, life satisfaction, negative social interactions, age, and identification of a primary source of stress were the most important. This study provides evidence that important factors associated with psychological stress come from various socio-ecological domains. However, prospective studies should be conducted to assess the causal directionality of these findings.

## Supporting information

**S1 Table. Detailed variable description.** A list of variables from the CCHS that were used in this analysis, including the name of variable, original wording (for items asked in the CCHS-MH) or concept (for derived variables), and the scoring used for the present study.
(DOCX)

**S2 Table. Weighted sample demographic information.**
(DOCX)

**S3 Table. Missingness of data and complete analysis results.** Missingness of each variable and primary random forest analysis, in terms of both raw (Imp.) and relative (Rel. Imp.) importance.
(DOCX)

**S4 Table. Sensitivity analyses.** The raw importance (Imp), relative importance (RI), and rank within each analysis (Rank) of each sensitivity analysis. Variables are listed in the order of importance in the original analysis with the complete dataset; rank denotes the level of importance in each sensitivity analysis. The Age (20–75) analysis includes only those between the ages of 20 and 75; this omits those over age 75, who were not asked the employment question. The No Coping analysis omitted two coping items (i.e., Coping - social support, coping skill), which were not asked of those who reported no primary stressor. Finally, the No Mental Illness analysis omits those who reported having mental illness including: depression, bipolar disorder, any anxiety disorder, PTSD, suicidal thoughts, mania, and hypomania. Negative social interactions, life satisfaction (R), and Age (R) were consistently seen to be three of the most important variables.
(DOCX)

**S1 Figure. Relative importance of all factors.** (R) denotes variables that are reverse coded.
(TIFF)

**S1 File. Regression results.**
(DOCX)

## Author contributions

**Conceptualization:** Benjamin A. Hives, Mark R. Beauchamp, Yan Liu, Eli Puterman.

**Formal analysis:** Benjamin A. Hives.

**Methodology:** Benjamin A. Hives, Eli Puterman.

**Project administration:** Benjamin A. Hives.

**Software:** Benjamin A. Hives, Yan Liu, Jordan Weiss.

**Supervision:** Mark R. Beauchamp, Yan Liu, Eli Puterman.

**Visualization:** Benjamin A. Hives.

**Writing – original draft:** Benjamin A. Hives.

**Writing – review & editing:** Benjamin A. Hives, Mark R. Beauchamp, Yan Liu, Jordan Weiss, Eli Puterman.

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
