## [Decision Letter · Decision Letter 0]

7 Jan 2025

PONE-D-24-46724Multidimensional correlates of psychological stress: Insights from traditional statistical approaches and machine learning using a nationally representative Canadian samplePLOS ONE

Dear Dr. Hives,

Thank you for submitting your manuscript to PLOS ONE. After careful consideration, we feel that it has merit but does not fully meet PLOS ONE’s publication criteria as it currently stands. Therefore, we invite you to submit a revised version of the manuscript that addresses the points raised during the review process.

We look forward to receiving your revised manuscript.

Kind regards,

Ali B. Mahmoud, Ph.D.

Academic Editor

PLOS ONE

Journal Requirements:

2. We noted in your submission details that a portion of your manuscript may have been presented or published elsewhere. [This manuscript is a revision of my thesis which was posted in UBC's public repository.] Please clarify whether this [conference proceeding or publication] was peer-reviewed and formally published. If this work was previously peer-reviewed and published, in the cover letter please provide the reason that this work does not constitute dual publication and should be included in the current manuscript.

4. Thank you for uploading your study's underlying data set. Unfortunately, the repository you have noted in your Data Availability statement does not qualify as an acceptable data repository according to PLOS's standards.

5. We notice that your supplementary figures are uploaded with the file type 'Other'. Please amend the file type to 'Supporting Information'. Please ensure that each Supporting Information file has a legend listed in the manuscript after the references list.

Reviewers' comments:

Reviewer's Responses to Questions

**Comments to the Author**

1. Is the manuscript technically sound, and do the data support the conclusions?

Reviewer #1: Partly

2. Has the statistical analysis been performed appropriately and rigorously? 

Reviewer #1: No

3. Have the authors made all data underlying the findings in their manuscript fully available?

Reviewer #1: No

4. Is the manuscript presented in an intelligible fashion and written in standard English?

Reviewer #1: Yes

5. Review Comments to the Author

Reviewer #1: The manuscript explores the correlates of psychological stress using both traditional statistical methods and a random forest algorithm. While it tackles an important public health issue and uses innovative methods, there are significant concerns about the methodology, particularly the way the random forest model was applied and evaluated.

The study uses random forest as its main machine-learning method but doesn’t follow common practices in the field. For example, the dataset wasn’t split into training, validation, and testing subsets. On top of that, there’s no mention of cross-validation, which is crucial for ensuring that the model is reliable and generalizable. Cross-validation methods, like k-fold or leave-one-out, are standard in machine learning to prevent overfitting and provide accurate error estimates. Without these steps, it’s hard to have confidence in the model’s performance. Adding these procedures in future revisions would greatly improve the robustness of the results.

The manuscript also doesn’t include key metrics for evaluating the random forest model, such as out-of-bag error, mean squared error (MSE), root mean squared error (RMSE), or mean absolute error (MAE). These metrics are important for assessing how well the model performs and for supporting the choice of predictors in the follow-up analyses. It would also help to clarify if the variable importance measures came from out-of-bag samples, which is a standard and reliable practice for random forest models.

Another concern is the use of a single-item measure for psychological stress. Treating this as a continuous variable without enough evidence to support its validity might affect the modeling outcomes. This limitation should be addressed in the discussion section.

The study provides interesting insights into the factors linked to psychological stress, but the methodological gaps, especially around cross-validation and model evaluation, weaken the findings. Making these improvements in future revisions would boost the reliability and impact of the study.

6. PLOS authors have the option to publish the peer review history of their article (what does this mean? ). If published, this will include your full peer review and any attached files.

**Do you want your identity to be public for this peer review?** For information about this choice, including consent withdrawal, please see our Privacy Policy .

Reviewer #1: No

---

## [Author Response · Author response to Decision Letter 1]

18 Feb 2025

We have included additional formatting in the response letter to more clearly demonstrate the changes that have been made. Please see the attached letter.

---

## [Decision Letter · Decision Letter 1]

4 Apr 2025

Multidimensional correlates of psychological stress: Insights from traditional statistical approaches and machine learning using a nationally representative Canadian sample

PONE-D-24-46724R1

Dear Dr. Hives,

We’re pleased to inform you that your manuscript has been judged scientifically suitable for publication and will be formally accepted for publication once it meets all outstanding technical requirements.

Kind regards,

Ali B. Mahmoud, Ph.D.

Academic Editor

PLOS ONE

Additional Editor Comments (optional):

Reviewers' comments:

Reviewer's Responses to Questions

**Comments to the Author**

1. If the authors have adequately addressed your comments raised in a previous round of review and you feel that this manuscript is now acceptable for publication, you may indicate that here to bypass the “Comments to the Author” section, enter your conflict of interest statement in the “Confidential to Editor” section, and submit your "Accept" recommendation.

Reviewer #1: All comments have been addressed

2. Is the manuscript technically sound, and do the data support the conclusions?

Reviewer #1: Yes

3. Has the statistical analysis been performed appropriately and rigorously? 

Reviewer #1: Yes

4. Have the authors made all data underlying the findings in their manuscript fully available?

Reviewer #1: Yes

5. Is the manuscript presented in an intelligible fashion and written in standard English?

Reviewer #1: Yes

6. Review Comments to the Author

Reviewer #1: The authors have adequately addressed the methodological concerns. They conducted additional analyses following common practices in machine learning and provided more benchmark metrics for model evaluation.

7. PLOS authors have the option to publish the peer review history of their article (what does this mean? ). If published, this will include your full peer review and any attached files.

**Do you want your identity to be public for this peer review?** For information about this choice, including consent withdrawal, please see our Privacy Policy .

Reviewer #1: No

---

## [Editor Report · Acceptance letter]

PONE-D-24-46724R1

PLOS ONE

Dear Dr. Hives,

I'm pleased to inform you that your manuscript has been deemed suitable for publication in PLOS ONE. Congratulations! Your manuscript is now being handed over to our production team.

Kind regards,

on behalf of

Dr. Ali B. Mahmoud

Academic Editor

PLOS ONE